

# Associative bond swaps in molecular dynamics

**Simone Ciarella[1,2]⋆ and Wouter G. Ellenbroek[2,3]†**

**1** Laboratoire de Physique de l'Ecole Normale Supérieure, ENS, Université PSL, CNRS,
Sorbonne Université, Université de Paris, F-75005 Paris, France
**2** Department of Applied Physics, Eindhoven University of Technology,
P.O. Box 513, NL-5600MB Eindhoven, The Netherlands
**3** Institute for Complex Molecular Systems, Eindhoven University of Technology,
P.O. Box 513, NL-5600MB Eindhoven, The Netherlands

⋆ simoneciarella@gmail.com, † w.g.ellenbroek@tue.nl

## Abstract

We implement a three-body potential to model associative bond swaps, and release it as part of the HOOMD-blue software. The use of a three-body potential to model swaps has been proven to be effective and has recently provided useful insights into the mechanics and dynamics of adaptive network materials such as vitrimers. It is elegant because it can be used in plain molecular dynamics simulations without the need for topology-altering Monte Carlo steps, and naturally represents typical physical features such as slip-bond behavior. It is easily tunable with a single parameter to control the average swap rate. Here, we show how associative bond swaps can be used to speed up the equilibration of systems that self-assemble by avoiding traps and pitfalls, corresponding to long-lived metastable configurations. Our results demonstrate the possibilities of these swaps not only for modeling systems that are associative by nature, but also for increasing simulation efficiency in other systems that are modellable in HOOMD-blue.



# 1   Introduction

The concept of a smart material capable of changing its properties in response to certain external queues is the foundation of many lines of modern research. Numerical studies and simulation have always been powerful in predicting material properties from their elemental building blocks [1]. In the context of smart plastics, vitrimers have recently taken the spotlight [2–6]. They are a new class of polymer networks that are as malleable and recyclable as thermoplastics while retaining the strength and resilience of thermosets. This unique combination of properties is provided by a chemical mechanism that makes covalent cross-links dynamic. The resulting bond exchange mechanism is connectivity-preserving, by virtue of being associative: the new partner moiety binds before the old one unbinds, thus preserving the total number of bonds. At low swap-rates, vitrimers behave like thermosets, while at high rates, they become malleable like thermoplastics. Going across this transition, bond-swaps make it possible to release internal stresses without losing the overall shape in unprecedented ways [5]. Interestingly, even DNA-based systems [7] can be made smart using a similar bond-swap mechanism [8].

Looking beyond vitrimers, the concept of having bonds that are long-lived and hard to break while being fully exchangeable can be used to improve the self-assembly of complex structures. The process of self-assembly, in which particles arrange without outside guidance, is a key concept in chemistry [9], biology [10], nanotechnology [11] and it is the foundation of complex computer simulations [12]. Self-assembly in computer simulations is tricky because it requires the binding energy to be large enough to create long-lived structures, which in turn also stabilize undesired metastable configurations, thus preventing the system from reaching equilibrium. Those metastable configurations can be seen as traps or pitfalls of the self-assembly process, because they increase the time required to reach equilibrium.

In this paper, we present the implementational details of a recently introduced method that provides associative bond-swapping in molecular dynamics (MD) simulations. We then use this method to demonstrate the use of this method in improving self-assembly of a simple colloidal model system.

Several numerical solutions for bond swaps have been developed in recent years, usually as Monte Carlo swaps in hybrid molecular dynamics or in fully Monte Carlo-based simulations [13–18]. The method that we propose and share here is an implementation of a fully MD-based method introduced in Ref. [19]. This recipe to model swaps has already been able to provide meaningful results in the context of smart vitrimers [20–24], or in the assembly of soft particles [25–27].

The method extends any pairwise potential able to generate network structures, making its bonds swappable by introducing a continuous three-body interaction term based on that same pairwise potential. This three-body addition is not only elegant and smooth, but also relatively cheap because it does not introduce any independent function that has to be computed every step: it only combines forces already evaluated by standard pairwise MD. An additional parameter ($\lambda$) that controls the swap rate through an energy barrier is introduced in the def-

inition of this three body potential, and it acts as the knob to tune the mechanical properties of the material through swaps.

The paper is organized as follows. We first present our implementation of the three-body potential for swappable bonds into the framework of the HOOMD-blue toolkit [28, 29]. We then demonstrate its effectiveness through a bond autocorrelation analysis, proving that bonds rearrange by swapping. We continue with a novel use case for the method, demonstrating the improvement and acceleration of a model self-assembling system by avoiding pitfalls through associative swapping of the assembling bonds. We demonstrate that, compared to self-assembly based on simple pair potentials, we reach better configurations in a shorter time. Finally, we demonstrate the primary physical advantage of the method: Because the swapping is governed by forces, the simulation follows the free energy landscape in deciding which particle have to swap. In this sense, the method provides a realistic network dynamics that captures practical effects like the slip-bond behavior discussed in sec. 4.2.

Our implementation in HOOMD-blue is available via the official repository, starting from version v3.0.0-beta.4, released on February 16, 2021. The potential can run both on CPU and GPU, both NVIDIA and AMD architectures.

## 2   The swap potential

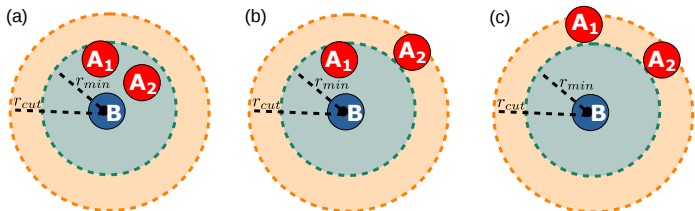

Figure 1: Depiction of three particles interacting that can form and swap A-B bonds. There are three different scenarios in which the three body potential is active. In (a) both A particles are within $r_m$ from B, so it follows from eqn (1) that the three body term is constant. In (b) only one particle is within $r_m$ causing the other to feel a repulsion from $B$ due to eqn (4). In (c) both $A_1$ and $A_2$ are beyond $r_m$ so they both feel a three-body force.

The *associative* bond swap scenario requires that the only mechanism to rearrange the bonds are in fact the swaps. This means that each bond has to be unbreakable by thermal fluctuations. Furthermore, the full potential needs to guarantee that each reversible binding moiety only binds to a single partner, to represent the fact that the bonding in the chemical system is 1-to-1 and does not clusterize. We call this the single-bond-per-site condition. The three-body mechanism accounts for all of these requirements [19] if we build it starting from a strong and short-ranged potential. Our choice, and the one we implemented in HOOMD-blue, is built upon a generalized Lennard-Jones

$$v_{ij}\left(\vec{r}_{ij}\right) = 4\epsilon\left[\left(\frac{\sigma}{r_{ij}}\right)^{2n} - \left(\frac{\sigma}{r_{ij}}\right)^{n}\right] \qquad r < r_{\text{cut}} , \tag{1}$$

which has a minimum of depth $\epsilon$ at a bond equilibrium distance of $r_{min} = \sigma\, 2^{1/n}$. The choice of $\epsilon = 100k_{\text{B}}T$ and $n = 10$ that we use in Refs. [20, 22, 27] guarantees short-range bonds that cannot be broken, well suited to mimic covalent-like bonding.

Then, the 3-body term is defined by how much the interaction between particles $i$ and $j$ is affected by the presence of other particles $k$ that are within range of particle $i$,

$$v_{ijk}^{(3b)} = \lambda \epsilon \, \hat{v}_{ij}^{(2b)}\left(\vec{r}_{ij}\right) \cdot \hat{v}_{ik}^{(2b)}\left(\vec{r}_{ik}\right) \, . \tag{2}$$

Thus, it consists of a product of two similar terms, each of which is derived from the two body potential as

$$\hat{v}_{ij}^{(2b)}\left(\vec{r}_{ij}\right) = \begin{cases} 1 & r \leq r_{\min} \, , \\ -\dfrac{v_{ij}\left(\vec{r}_{ij}\right)}{\epsilon} & r > r_{\min} \, . \end{cases} \tag{3}$$

We have also introduced the three body energy parameter $\lambda \geq 1$ that has the role of tuning the energy barrier for a swap event. In HOOMD-blue the class *md.pair.revcross* invokes eqns (1,2), where the parameters can be specified using *pair_coeff.set([types],[types],sigma,n,epsilon,lambda3)* as explained in the official HOOMD-blue documentation.

Since MD is based on the solution of Newton's equations of motion, we need to derive the three-body force acting on the particles involved in a swap. Suppose that the interaction of eqn (1) is only defined between particle of type A and type B (respectively red and blue in Fig.1). A swap event can happen if two particles $A_1$ and $A_2$ are within the cutoff distance $r_c$ from B, such that they are interacting. We distinguish 3 possible scenarios depicted in Fig. 1 related to the action of the three body potential of eqn (2). If both $A_1$ and $A_2$ are within $r_{\min}$ then $v_{BA_1A_2}^{(3b)} = const.$ and thus the three body potential does not provide any force (its derivative is zero). Due to thermal motion $A_2$ might move farther than $r_{\min}$. In this situation (b) we have that:

$$v_{BA_1A_2}^{(3b)} = \lambda \epsilon \, \hat{v}_{BA_2}^{(2b)}\left(\vec{r}_{BA_2}\right) = -\lambda v_{BA_2}\left(\vec{r}_{BA_2}\right) \, , \tag{4}$$

thus only $A_2$ would feel a force. In eqn (4) the role of the parameter $\lambda$ is clearly visible: (i) if $\lambda = 1$ then the three body term in eqn (4) exactly shields the attraction between $A_2$ and $B$ without influencing the $A_1 - B$ bond, (ii) if instead $\lambda > 1$ the contribution from eqn (4) beats the $A_2 - B$ attraction making it harder for $A_2$ to get closer to $B$ and "steal" the bond from $A_1$. This effectively defines a swap energy barrier $\beta \Delta E_{sw} = \beta \epsilon (\lambda - 1)$ that grows linearly with $\lambda$. Lastly (iii), if $\lambda < 1$ then eqn (4) is not enough to compensate the attraction and the system will form both $A_1 - B$ and $A_2 - B$ going toward full clusterization around the swapping groups. Moreover, if more than three particles are in the interaction range, terms like eqn (4) will strongly suppress the attraction (if $\lambda \geq 1$), providing the single-bond-per-site condition. Finally in Fig. 1(c) both $A_1$ and $A_2$ are above $r_{\min}$ so they both feel an effect due to eqn (2) that will allow only one of the two to get within $r_{\min}$ from $B$.

In Fig. 2 we summarize the energy changes while undergoing a swap event. The energy from the formation of the second bond is compensated by the three-body term producing an overall flat energy landscape that allows $A_1$ to steal the bond from $A_2$ without breaking it first.

## 2.1 Pressure

The three body term is non-zero only for transient states while a bond is swapping. Still, those transient states have to be considered while evaluating thermodynamic quantities, because they characterize the system and become more and more common as the density increases. For this reason we have included in the pressure calculation automatically preformed by HOOMD-Blue the element of the pressure tensor that come from triplet interacting via eq. 2. In the Supplemental Material of Ref. [20] we show how to derive them from the standard virial approach [30]. Those calculations are quite tractable because our three body potential is

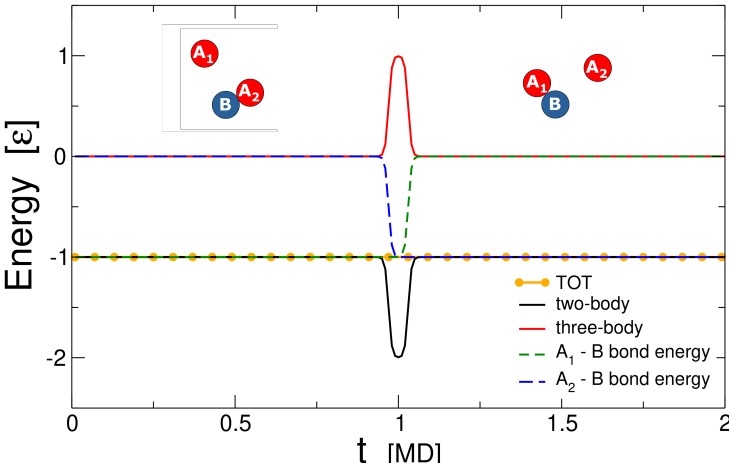

Figure 2: Time dependence of different components of the energy along a swap event at $t = 1$. When the particle $A_1$ gets closer to $B$, its two-body energy decreases (blue dashed line), but this change is compensated by the three-body term (red). The triplet state is short lived, in fact particle $A_2$ leaves quickly after the formation of the triplet. Noticeably the total energy (yellow) stays always constant.

actually a combination of two body terms, so it is possible to take its derivative and get the following virial-like expression:

$$\sigma_{\alpha\beta} = -P^{(2b)}_{\alpha\beta,\text{virial}} - \frac{\lambda\epsilon}{V} \sum_{ijk}^{*} \left[ \vec{F}_{ij}(r_{ij})_\alpha \left(\vec{r}_{ij}\right)_\beta \, \hat{v}^{(2b)}_{ik} + \hat{v}^{(2b)}_{ij}\vec{F}_{ik}(r_{ik})_\alpha \left(\vec{r}_{ik}\right)_\beta \right], \tag{5}$$

where $F$ is the force coming from the potential defined in eqn (3). We can use this tensor for any thermodynamic measurement, even the stress relaxation modulus if we use the autocorrelation method [13, 31, 32]

$$G(t) \approx \frac{V}{k_\text{B}T} \left\langle \overline{\sigma_{\alpha\beta}(t)\sigma_{\alpha\beta}(0)} \right\rangle . \tag{6}$$

This is important because stress relaxation is a crucial feature of dynamic networks.

## 3 Results

In this work we describe two applications of the three-body method for associative bond swaps. The first one involves smart vitrimeric materials that use bond swaps to make their network structure dynamic. Using a dumbbell-forming mixture, based on the potential introduced in sec. 2, we measure the dynamic effect of bond swaps at equilibrium. We use the same model for a performance assessment and quantify the speed up provided by the use of GPUs, that is reported in the Supplementary Information. Secondly, we discuss the application of bond swaps in self-assembly of network-forming patchy colloids.

### 3.1 Effect of swaps

To test our implementation in the context of bond swapping materials, we compare our results with Ref. [19] by setting $n = 100$ in eqn (1). Furthermore we model the $AA$ and $BB$ interactions as repulsive Weeks-Chandler-Andersen (WCA) potentials [33] with $\sigma_{WCA} = \epsilon_{WCA} = 1$. The number density is set to $\rho\sigma^3 = 0.125$ while the temperature is $k_B T/\epsilon = 0.03$. In this condition

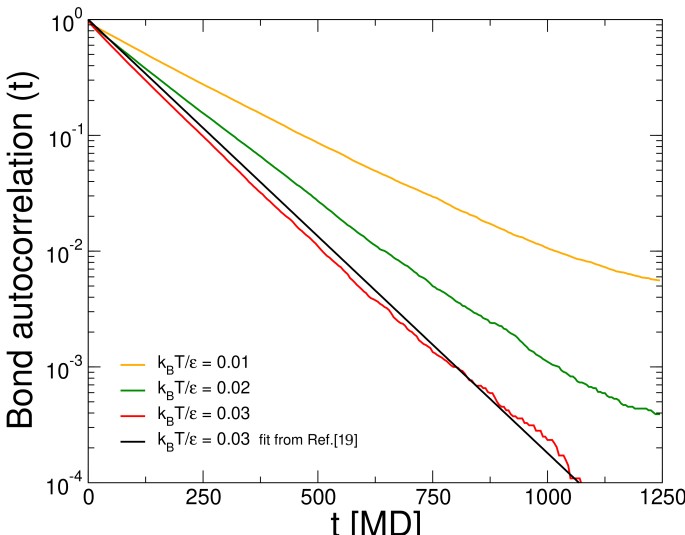

Figure 3: Bond autocorrelation function for a binary mixture of WCA particles with the additional three body potential in eqn (2). Its decay is exponential and the characteristic time depends on the temperature. Here we show that our HOOMD-blue implementation is compatible with Ref. [19].

a mixture of $N_A = 600$ particles of type $A$ and $N_B = 400$ $B$-type forms $N_B$ dumbbells because all the minority $B$ particles are always bonded. Nevertheless they can swap $A$ partners through bond-swaps with the reservoir of $N_A - N_B$ unbonded particles. To quantify this mechanism we measure the bond autocorrelation function in Fig. 3. This quantity corresponds to the fraction of bonds present at time 0 that are still unswapped at time $t$. Its decay is then a proof of the effectiveness of bond swaps, since bond breaking is prevented by the low temperature. In Fig. 3 we show that our results are compatible with Ref. [19], while also showing that the relaxation time depends on the temperature because, for higher values of $T$, the particles move faster so they are more likely to bump into each other and swap.

## 3.2 Self-assembly pitfalls

We demonstrate the use of our bond-swap method to avoid long-lived metastable states in self-assembly simulations, thus providing a way to anneal self-assembling systems numerically. To this end, we simulate in HOOMD-blue a twodimensional mixture of $N_t = 400$ trifunctional and $N_d = 600$ difunctional monomers. The ratio $N_t/N_d = 2/3$ is chosen such that all the particles can be fully bonded. Each particle is a WCA repulsive disk ($\sigma_{WCA} = \epsilon_{WCA} = 1$) with two or three attractive patches. We compare self-assembly simulations between a swapping system and a reference system using only pair potentials for the patches. In the swapping system the patches are represented by the three-body potential, following eqns (1,2). In the reference system the patches have a gaussian attraction:

$$V_{\text{patch}} = -\epsilon_{\text{p}} \exp\left[ -\frac{1}{2}\left(\frac{r}{\sigma_{\text{p}}}\right)^2 \right] \qquad r < r_{\text{cut}} , \qquad (7)$$

where we let the attraction strength $\epsilon_{\text{p}}$ vary. The attraction length $\sigma_{\text{p}} = 0.1$ geometrically imposes 1-to-1 bonds [34]. We define two patches bonded if they are closer than $d_{\text{bond}} = 2\sigma_{\text{p}} = 0.2\sigma_{\text{WCA}}$. We do canonical Langevin dynamics at $k_B T/\epsilon = 1$ in 2d square boxes of side $L = 39.63\,\sigma_{WCA}$. with periodic boundary conditions. We use a timestep of $dt = 10^{-3}$ and we average $M = 10$ independent realizations.

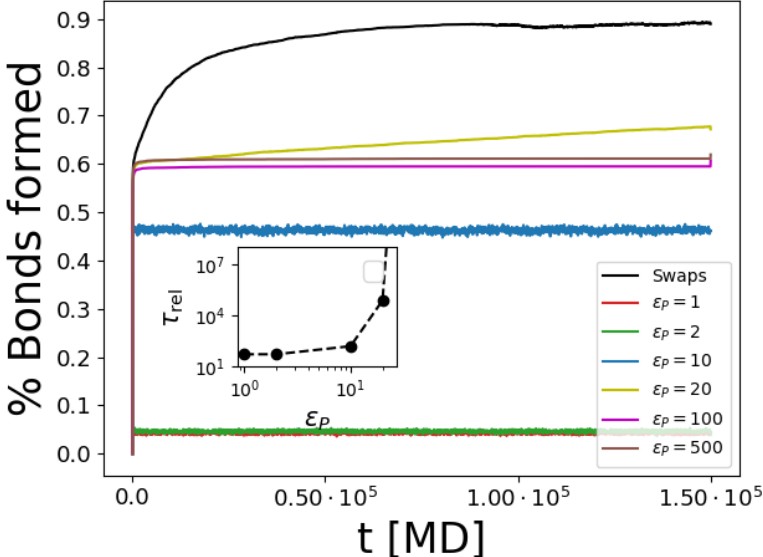

Figure 4: Percentage of bonds formed by the system introduced in sec. 3.2. The final stage of self-assembly corresponds to 100% bond formation. The non-swapping systems are trapped in pitfalls, so they are hardly forming new bonds. In the inset we show the characteristic time of the bond autocorrelation function $\tau_{\mathrm{rel}}$. When bonds are weak ($\epsilon_p < 10$) only few bonds occur at the same time. On the other hand, if the bonds are strong ($\epsilon_p > 10$) the system cannot escape pitfalls because defects are long-lived. Bond swaps can be used to have strong long-lived bonds, which at the same time can relax away defects.

At low temperatures where the binding energy dominates thermal fluctuations, the equilibrium states that minimize the free energy for the system are the fully bonded networks, because the binding energy is the driving term. We are interested in understanding which simulation protocol is the optimal to reach that. As a consequence in Fig. 4 we asses how close the system is to the target equilibrium by counting the number of bonds at time $t$, normalized with the maximum number of bonds allowed by the mixture $N_{\mathrm{max}} = 2N_d$. Furthermore, we report in Fig. 5 a snapshot for the different simulation protocols, to visualize equilibrium states and relative pitfalls.

We compare different binding energy $\epsilon_p$ to bond swaps. In the inset of Fig. 4 we report the relaxation time $\tau_{\mathrm{rel}}$ of the bond autocorrelation function (sec. 3.1). The steep growth of $\tau_{\mathrm{rel}}$ proves that if $\epsilon_p > 10[k_B T]$ the bonds are very long-lived. As a consequence the simulations with strong bonds do fall into pitfalls corresponding to configurations where a difunctional bead is connected with another difunctional one. In fact, we see in Fig. 5 that for $\epsilon_p = 500$ the network is composed by long (red) branches, resulting from bonding between difunctional beads, that act as pitfalls limiting the reservoir of free difunctionals available for the trifunctional (blue) particles. Due to the significant energy of each bond, those pitfalls are very long-lived. As a result, strong bonds do not assemble beyond 70%. On the other hand, a small $\epsilon_p$ prevents pitfalls but at the same time it only forms very few bonds simultaneously and thus it can never reach the fully bonded network. We see this in Fig. 5 for $\epsilon_p = 1$, where hardly any bond is formed. The best solution to form the maximum amount of bonds is then to use swaps. With strong bonds that can also swap, the system can escape the pitfalls by swapping bad configurations with good ones and go towards the fully bonded network. This is confirmed in Fig. 4, where the swapping system (black) keeps forming new bonds, surpassing any standard simulations where bonds do not swap. The effect of swaps is also evident in

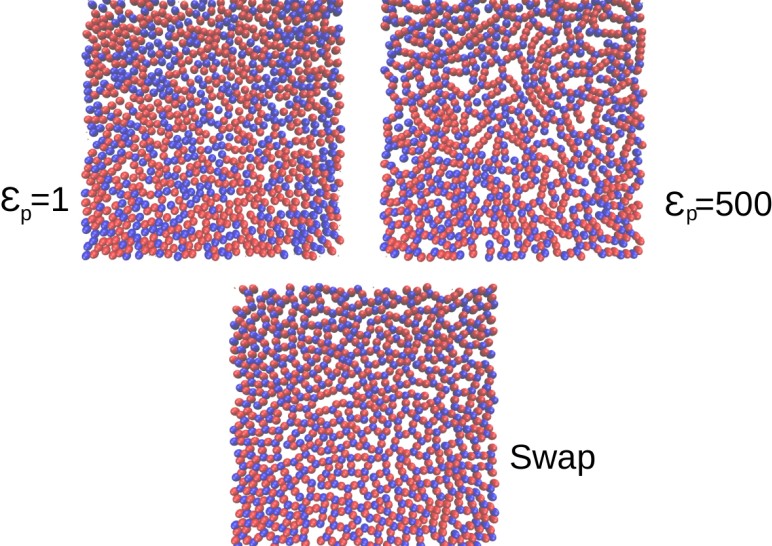

Figure 5: Self-assembly after $10^7$ timesteps with different binding interactions. For weak bonds ($\epsilon_p = 1$) the mixture of trifunctional (blue) and difunctional (red) beads hardly forms any persistent bond. When the bonds are strong ($\epsilon_p = 500$) they are also very long-lived and a network with long branches is assembled. There are in fact long sequences of bonded red beads that should not be present at equilibrium. Lastly, for swapping moieties (swap), the self-assembly goes even further producing a network with shorter branches and fewer open endings.

Fig. 5, where the network that self-assembled using swaps has fewer open branches, thus being the only one to approach the fully assembled state.

## 4 Discussion

### 4.1 Additional validation and performance

In the supplementary material, we demonstrate that spatial distribution of swap events in a homogeneously prepared system is itself also homogeneous. This means homogeneous systems are stable to the use of our swap method, and that it is for example not the case that a few swaps in a certain region will locally increase the swap rate in that region, which could drive the system unstable.

The supplementary material also contains a section on performance of the swap method, comparing the CPU and GPU implementations in HOOMD-blue, where we find that, as in the existing parts of the HOOMD-blue project, the speedup obtained with the GPU becomes significant for larger systems ($N > 8192$).

### 4.2 Physical advantages of the method

The 3-body bond-swapping method captures some nontrivial physical properties of adaptive polymer networks. We get these additional effects *for free* in the sense that they arise simply from the fact that the method is based on potentials and forces.

A first example is demonstrated through the effect of setting the mass of half of the *A* moieties to 1/10. In this situation, a legitimate algorithm to model swaps would then bias the lighter *A* particles to swap more, because they have a higher thermal velocity and there-

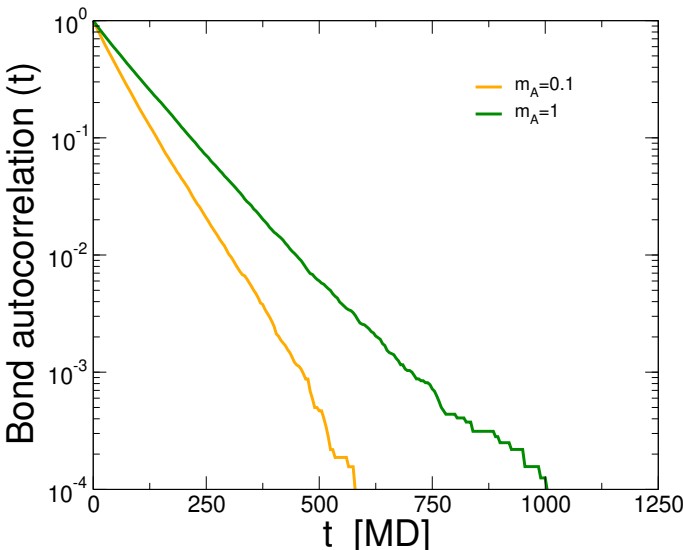

Figure 6: Comparison of the bond autocorrelation function evaluated for the lighter $A$ particles bond (orange) and for the heavier (green). Data refer to $k_B T / \epsilon = 0.03$. The characteristic time goes from 98 for the heavier, to 60 for the lighter. Notice that both of them decreased from the original value of 116 in Fig. 3, because now even the heavier particles have more bumps due to the faster light particles.

fore are more likely to be the first to escape out of the three-body intermediate state. This is indeed what we confirm for our implementation in Fig. 6, where we compare the bond auto-correlation functions for the bonds with the lighter (orange) and heavier (green) $A$ particles. Our algorithm can capture this effect because eqns (1,2) authentically explore the free energy landscape of the system, without requiring any external forcing to favor the swap of the lighter moieties while relying only on enthalpy and entropy.

The principle behind this feature should be expected to work more broadly: As a second example, bonds that are under a significant tensile force will also swap more easily: The pulling force aids in deciding which $A$-particle gets away, as one would expect in any simple slip-bond.

Thus, the physically expected effects on swap rates of parameters like mass and tension are build-in in our model, in contrast to hybrid models in which every dependence needs to be put in by hand.

### 4.3 Choosing the species concentrations

Typical applications of the method involve an energy scale $\epsilon$ for the three-body potential that is large enough that nearly all possible bonds will form and the three-body swap is the only mechanism for bond exchange. Again calling the majority species $A$ and the minority species $B$, this implies the concentration of bonded A-B pairs is equal to $c_B$ and the concentration of free beads of type A is equal to $c_A - c_B$. In a mean-field rate equation approach, ignoring any spatial correlations and differences in diffusivities between species, this gives an esitimate for the swap rate as a function of species concentrations as

$$k_{\text{swap}} \propto c_B (c_A - c_B) . \tag{8}$$

The picture is that a swap involves an encounter of a free A-bead with a bonded pair so the rate should be proportional to the product of the two concentrations.

Analyzing this expression at fixed $c_B$ (so at fixed total number of bonds) gives the rather obvious result that adding more A-type beads will always increase the swap rate. A more

interesting observation is obtained for the case in which the total concentration of reactive beads is kept fixed. Optimizing the swap rate estimate in Eq. (8) under the constraint that $c_A + c_B = c$ gives $c_A = \frac{3}{4}c$ and $c_B = \frac{1}{4}c$. This result can be used as a first-order guess for the composition of a system if having a large swap rate is a goal.

### 4.4 Applications to vitrimers

Studying the dynamics of vitrimers was our original motivation for implementing this method and we believe there is still more to be done in this area. The method can be added to any coarse-grained polymer model, and is therefore suited to address the types of questions generally answered with coarse-grained models. In the context of vitrimers, this includes relations between polymer architecture and network dynamics, and between network dynamics and macroscopic behavior such as rheology [20] and self-healing [22]. The details of the coarse-graining are similar to the case of dissociative swapping with simple potentials or patchy interactions, with of course the addition of having to choose the parameters (in particular the energy barrier) of the three-body potential in order to capture the chemistry of swapping as closely as possible. This includes the choices of relevant time scales: When these are based on the self-diffusion time of a monomer, the addition of the three-body beads will not have a major effect on the choice of time scales in the coarse-graining procedure.

## 5 Conclusion

In this paper we show that our HOOMD-blue implementation of the bond swap algorithm using a three-body potential provides associative bond-swapping in dynamic network simulations with a tunable energy barrier. The method, originally introduced in Ref. [19], can be now used for simulations of any network materials that feature chemical moieties that enable associative swapping, such as vitrimers. In addition, the bond-swapping provides a way to speed up simulations of self-assembling systems with interactions that are so strong that traditional simulations would spend prohibitively long times in metastable states. In these systems it provides a shortcut to both have strong bonds and a way to effectively anneal the system without having to play with delicate changes in temperature. The most important feature of the method is that it is based on potentials only, and therefore suited for molecular dynamics simulations, allowing to study the proper dynamics of network materials. Capturing the dynamics of adaptive network materials correctly is key for simulations that aim to unravel their mechanical properties. The tunability of the three-body potential provides an accessible parameter to control the swap rate and thus the macroscopic properties of the modelled material. The method is efficient in the sense that, even though it does involve the evaluation of forces arising from a three-body potential, those forces can be calculated in terms of two-body forces that had to be calculated anyway, and therefore the three-body forces are relatively cheap. Its efficiency makes it such that any network forming system might benefit from its use.

Lastly we show that the algorithm intrinsically captures physical effects of parameters affecting the swap rates, like the mass of the swapping moieties, which would have to be added by hand in hybrid implementations involving topology-altering Monte Carlo steps. We hope that our HOOMD-blue implementation will be of help for anyone interested in efficient network assembly or dynamical properties of smart and adaptive materials.

## Conflicts of interest

There are no conflicts to declare.

## Data availability

The data and the codes to reproduce the results of this paper are available at: gitlab.tue.nl/SCiarella/Associative_bond_swaps_in _MD.

## Acknowledgements

We are grateful to Joshua Anderson and the HOOMD-blue development team for helping us create a more compatible and maintainable code. We also thank Francesco Sciortino for the support and useful discussions.

## Supplementary Information

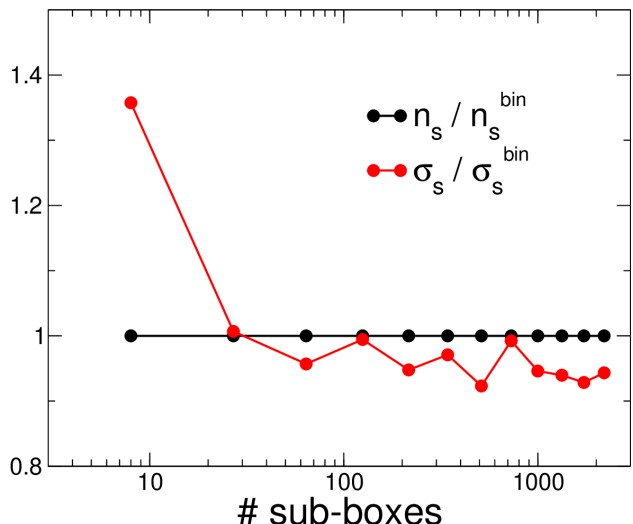

Figure 7: Average number of swaps $n_s$ measured by dividing the simulations of sec. 3.1 into sub-boxes, and their standard deviation $\sigma_s$. We compare the numbers calculated in the simulations to values expected from the binomial distributions (bin). The similarity between those quantities proves that swaps events are homogeneously distributed in space.

### Validation: Spatial homogeneity of swaps

In this section we show that the algorithm is *genuine* in the sense that it captures some of the physical mechanisms of the system without requiring any additional information, at least for the simple models we tested. First, we study the locations at which swap events happen in a simulation that is set up to be homogeneous, and verify that they are homogeneously distributed throughout the system. We start from the model system introduced in Sec. 3.1. Here, the two moieties have the same shape, mass and interaction potentials, so this system should be uniform. To test this, we pinpoint the locations where each swap took place in the $k_B T/\epsilon = 0.03$ simulation. To check the homogeneity, we divide the simulations into sub-boxes of the same size and we check the average number of swaps $n_s$ and its standard deviation $\sigma_s$. In Fig. 7 we compare those quantities with expected values from the binomial distribution. Results in Fig. 7 show that swaps are homogeneusly distributed. It follows that the algorithm is capable of capturing homogeneous systems.

**Performance: CPU vs GPU optimization**

One of the strengths of this three-body potential is its relative cheapness. In fact the definition in eqn (2) is based on two body terms, already evaluated by the standard MD routine. The largest computational price is then the accounting of triplets in the iteration procedure but not the three-body function itself. For this reason we try to reduce as much as possible the cutoff radius, going from 1.3 to 1.15. Notice that for $n = 100$ the value at the cutoff $v_{ij}(1.15) \approx 3 \cdot 10^{-4}$ is 4 orders of magnitude smaller than the standard Lennard-Jones potential at the typical cutoff of $2.5\sigma$. Additionaly, a significant speedup can be achieved using the GPU version of the RevCross potential. In Fig.8 we compare the number of timesteps executed in a second on a single Intel Xeon E5-2550 CPU and on a Tesla v100, for two different values of the cutoff. We see that already for $N = 2^{10}$ particles the GPU acceleration provides a speed up factor of $\approx 2$. More interestingly, this speedup factor drastically increases with the system size, reaching up to a factor 30 after $N = 2^{14}$.

Lastly, when the RevCross potential is used to model only the active group of a larger molecule as in Ref. [20,22,23,27] the simulations are even faster because only a finite number of components invoke three-particles neighbour lists and they benefit even more of the GPU acceleration. In particular, the simulations in Ref. [20] benefited from a speed up factor of 50 when evaluated on a Tesla P100-PCIE-16GB GPU. It follows that a more complex model that aims to include a sub-set of swapping moieties can then capitalize on the full speed up factor provided by the GPU implementation, which is usually larger than 15 [28] .

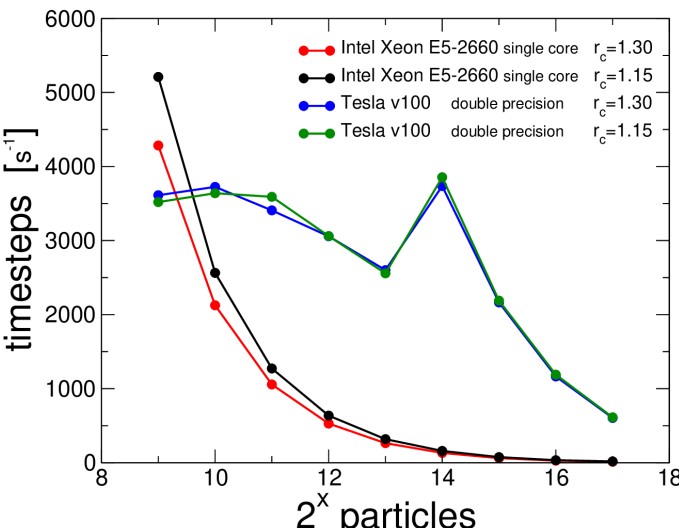

Figure 8: Computational time required by our HOOMD-blue implementation of the three-body swap potential. We test mixtures of $N = 2^x$ dumbbells introduced in the paper, ranging from $x = 9\,(N = 512)$ to $x = 17\,(N = 131072)$. We also test the effect of a reduction of the cutoff radius $r_c$ that limits the number of interacting triplets. Only for very small system size $(N < 1000)$ the CPU (Intel Xeon E5-2660) is faster than the GPU (Tesla v100). In this worst case scenario of a system where every particle interacts with the three body potential, as soon as $N > 1000$ the GPU becomes almost two times faster than the CPU. This speed up factor provided by the GPU grows for larger systems, reaching and surpassing a factor of $\sim 30$ at $N \sim 15000$, confirming that the GPU architecture is optimal for larger systems that capitalize on parallel architecture.

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
