# Peer review of "Associative bond swaps in molecular dynamics"

_SciPost Physics, doi:SciPost Phys. 12, 128 (2022)_

## Round 3 · Referee Report · Anonymous (Referee 1) · 2021-12-22

Strengths

  • Versatile algorithm
  • Highly reproducible

Report

Algorithms for bond swapping have been recently introduced in both Monte Carlo (MC) and Molecular Dynamics (MD) simulations: they allow, e.g., for simulating self-healing and restructuring properties of network forming materials as well as for equilibrating associative systems faster. While in MC simulations a bond swap was implemented as a stochastic event, a possible -- quite efficient --implementation for MD simulations relies on the introduction of a three-body potential, which compensates the energy gain associated to the formation of a double bond.

This contribution presents (and provides in an official repository) an implementation of the three-body potential for swappable bonds into the framework of the HOOMD-blue toolkit.Two examples are provided as bed tests: 1. the same binary mixture of mutually attractive spheres used in the reference publication by F. Sciortino (reference 19 of this paper): the reproduction of the reference behavior is fully achieved; 2. a binary mixture of di- and tri-functional patchy colloids: the swap algorithm is proven to achieve an effective annealing towards to the ground state, thus guaranteeing a faster equilibration, compared to self-assembly based on simple pair potentials.

The contribution is significantly useful for many purposes within the area of soft matter simulations and I support its publication. Here below I drop some observations to possibly improve the quality of the manuscript.

Requested changes

  • As the aim of part 2 is to demonstrate numerical efficiency, estimates for three-dimensional systems would be interesting.
  • How is a bond between two patches defined? The gaussian attraction between them has tails...
  • Even though the proof of principle is in place, it would be interesting to see the full equilibration curve of the patchy colloid system --until the fully bonded state is reached (it should be doable as it is a system of 120 particles in two-dimensions)
  • Why is the box shape rhombic in part 1?
  • From the brown and purple curves in Figure 4, it seems the systems freeze almost instantaneously into the network state, is that so?
  • I naively assumed the 2body-interactions between patches would be described by (1) plus (3), while the 3body-interaction would be described by (2). But I am not completely sure about it because of the dangling sentence "Each particle is a WCA repulsive disk with two or three attractive patches represented by either a three-body potential like eqns (1,2) if swaps are used", where "either" suggests a missing "or". What is the two-body potential?
  • The reference to the use of the RevCross is only for insiders...
  • Error in the text (while it is correct in the caption of figure 6): lighter (green) and heavier (orange) A particles.
  • The difference between the networks shown in figure 5 is a bit qualitative and relies on a trained eye.

---

## Round 3 · Referee Report · Lorenzo Rovigatti (Referee 2) · 2021-12-28

Strengths

1- Well written, clear and to the point 2- This implementation will be very useful for simulations of network-forming systems

Weaknesses

1- It does not fulfill any expectations listed in the journal's "acceptance criteria" 2- the "pressure" subsection seems a bit disconnected from the rest of the paper 3- the quality of some of the figures may be improved

Report

The package developed by the authors seems a very useful addition to the toolset of a molecular simulator interested in network-forming systems, and as such I believe that the paper should be published. However, looking at the acceptance criteria listed on the SciPost Physics's website I cannot see this paper meeting any of the "expectations". Looking at the requirements I believe that a more appropriate journal for this paper would be SciPost Physics Codebases, and this is why I'm ticking the "Accept in alternative journal" box. Now about the paper content. The paper is clear and well-written, and I have only a few comments that the authors may find useful: - The "pressure" subsection seems a bit out of context. I understand the importance of computing the thermodynamics right, but it's not clear whether the relation they present is already included in their HOOMD package or if it's there just for reference in case someone needs to compute the pressure in their simulations. - Fig. 5 is a bit weird, as it looks like the authors simulated very small systems and used periodic-boundary conditions to draw larger images. If this is the case then it should be explicitly stated or (even better) the simulations should be re-run with a larger number of particles. Moreover, looking at fig. 4 it looks like even with the swap the system has not equilibrated yet at the end of the simulation. I think that the authors should pick a system which can get to the ground state (i.e. ~100% bonds formed) with the swap mechanism. - Fig. 3 has a similar issue in that the three curves are a bit noisy. If it is not too much work I'd ask the authors to run longer simulations to clean up the data. - Even though the authors explicitly write that the 3-body algorithm is not their own, there are a few places scattered in the paper where a distracted reader may get the wrong impression (see for instance the first sentence of IV.B or in V., "the most important feature of our method") - I think Fig. 7 may very well be included in the main text, since it is already introduced therein. -It would be nice to understand what is the impact of the 3-body part on the performance of the code. The authors state that using fewer beads interacting with this 3-body potential makes the code go faster, but some hard numbers would help an interested reader.

Requested changes

1- Add details about pressure calculation in systems with 3-body swaps in HOOMD 2- Perform simulations of larger systems that get to the equilibrium state with the swap for Figg. 4 and 5 3- Make sure that it is clear everywhere in the text that the authors are presenting an implementation of an already-existing algorithm 4- (optional) Improve the quality of the curves in Fig. 3 5- (optional) Add Fig. 7 to the main text 6- (optional) add some details/ballpark numbers about the impact of using a certain fraction of "swapping beads" in a simulation

---

## Round 3 · Referee Report · Anonymous (Referee 3) · 2022-1-2

Strengths

The implementation of the method is useful.

Weaknesses

Connections with experiments is not provided.

Report

Authors implemented a three-body potential used for simulating vitrimers in HOOMD-blue software. As the authors mention, the method does not require using an MC step to alter the topology. The analyses presented in the paper are convincing, and I find the implementation in the software very useful. The manuscript is written clearly and in an acceptable format.

Requested changes

(1) One area that needs to be addressed is the connection with experiments. Specifically, how should one use this simulation toolkit and get consistent rheological data as obtained by Leibler and coworkers?
(2) As a follow-up to the previous comment, what is the MD time here? Is this the dimensionless time based on the Lennard-Jones potential parameters? It would be helpful if the authors showed the equations of motion and compared the time units with the real-time.
(3) The reference cited in Figure 3 and its caption does not match.

---

## Round 4 · Referee Report · Anonymous (Referee 1) · 2022-2-3

Report

The journal criteria are met, the paper can be published as it is.

---

## Round 4 · Referee Report · Lorenzo Rovigatti (Referee 2) · 2022-2-11

Report

The authors have provided satisfactory answers to all the points I have raised in my report. I am therefore happy to support publication of the manuscript in its current form.

---

## Round 4 · Referee Report · Anonymous (Referee 3) · 2022-2-24

Report

Authors address all comments. I recommend publishing the paper as is.

---

## Round 4 · Author Response

Thank you for sending us the reports on our manuscript "Associative bond swaps in molecular dynamics". We appreciate your efforts to provide an in-depth review of our work.
All the three Reviewers expressed positive comments on the validity, significance, originality, clarity, formatting and grammar, as well as insightful comments to further improve the manuscript.
Following the Reviewers’ suggestions, we have performed a series of additional calculations and have revised the manuscript accordingly.
Below we address all of the Reviewer’s comments and questions, including the corresponding changes in the manuscript. We believe these changes have significantly improved the quality of our manuscript and we hope that with this revision our work will be suitable for publication in Scipost Physics.

---

## Round 4 · List of Changes

*Report 1:

(1.1) As the aim of part 2 is to demonstrate numerical efficiency, estimates for three-dimensional systems would be interesting.
- While the part about network self assembly is in 2d, the rest of the paper is about 3d dumbbells, so Figs. 3,6-8 already refer to 3d systems.

(1.2) How is a bond between two patches defined? The gaussian attraction between them has tails...
- That is true. We now explicitly say in page 4 that: We define two patches bonded if they are closer than d=2 sigma_p=0.2 sigma_WCA.

(1.3) Even though the proof of principle is in place, it would be interesting to see the full equilibration curve of the patchy colloid system --until the fully bonded state is reached (it should be doable as it is a system of 120 particles in two-dimensions)
- Following the suggestion of Ref.2, we decided to rerun a larger self-assembly system (from N=120 to N=1000). We stop the simulations when the swapping system forms more than 90\% of the bonds, which corresponds to ~5days of cpu time. The new figure 4 gives a much more complete picture of the equilibration process.

(1.3) Why is the box shape rhombic in part 1?
- There was no need for the rhombic shape and it was there for historic reasons. We decided to re-run new simulations in square boxes.

(1.4) From the brown and purple curves in Figure 4, it seems the systems freeze almost instantaneously into the network state, is that so?
- Yes the referee is correct. While the small size of the system in the old version of Fig.4 exacerbated the difference between strong and weaker bonds, it is still evident that most of the bonds of the non-swapping mixtures appear almost instantaneously on our timescale.
This strengthen our conclusion that a mechanism to rearrange bonds is extremely valuable in many circumstances.

(1.5) I naively assumed the 2body-interactions between patches would be described by (1) plus (3), while the 3body-interaction would be described by (2). But I am not completely sure about it because of the dangling sentence "Each particle is a WCA repulsive disk with two or three attractive patches represented by either a three-body potential like eqns (1,2) if swaps are used", where "either" suggests a missing "or". What is the two-body potential?
- We edited the sentence to clarify that the gaussian attraction we mention is the interaction that we use when modelling non-swappable systems.

(1.6) The reference to the use of the RevCross is only for insiders...
- The three-body potential discussed in this paper can be directly used from HOOMD-blue with the name 'RevCross', so we decided to mention this fact to make the paper more pedagogical for possible HOOMD-blue users.

(1.7) Error in the text (while it is correct in the caption of figure 6): lighter (green) and heavier (orange) A particles.
- Thank you for pointing this out. We fixed this.

(1.8) The difference between the networks shown in figure 5 is a bit qualitative and relies on a trained eye.
- We now highlight the differences a bit more in the caption where we say that: [for strong bonds] There are in fact long sequences of bonded red beads that should not be present at equilibrium.

*Report 2:

(2.1) Add details about pressure calculation in systems with 3-body swaps in HOOMD
- We now explicitly say that the 3-body pressure is automatically calculated by HOOMD, in the same way it can be done for any other potential.
For more details about the calculation we say that: " In the Supplemental Material of Ref. [20] we show how to derive them from the standard virial approach"

(2.2) Perform simulations of larger systems that get to the equilibrium state with the swap for Figg. 4 and 5
- Following the suggestion of the Referee we run simulations of N=1000 units to update Fig.4-5.

(2.3) Make sure that it is clear everywhere in the text that the authors are presenting an implementation of an already-existing algorithm
- We reworked several statements to make this point more clear.

(2.4) (optional) Improve the quality of the curves in Fig. 3
- We think the improvement in the figure that might be achievable would not be worth the effort required to realise it. Since the referee marked this suggestions as optional, we decided not to pursue it.

(2.5) (optional) Add Fig. 7 to the main text
- We prefer to keep this figure in the supplement.

(2.6) (optional) add some details/ballpark numbers about the impact of using a certain fraction of "swapping beads" in a simulation
- Thank you for this very nice suggestion. We added a sec.IV C to specifically discuss this.

*Report 3:

(3.1) One area that needs to be addressed is the connection with experiments. Specifically, how should one use this simulation toolkit and get consistent rheological data as obtained by Leibler and coworkers?
- We agree with this point since studying vitrimers was also our original motivation for developing the method. We added sec.IV D to discuss more details about how vitrimers can be modelled.

(3.2) As a follow-up to the previous comment, what is the MD time here? Is this the dimensionless time based on the Lennard-Jones potential parameters? It would be helpful if the authors showed the equations of motion and compared the time units with the real-time.
- In sec.IV D we also discuss about the MD timescale. In particular, we mention that the timescale does not depend on the three-body potential, but rather on the coarse-grain parameters that would also need to be chosen in a simulation that does not use the three-body approach.

(3.3) The reference cited in Figure 3 and its caption does not match.
- Fixed. Thank you.

---

## Editorial Decision

published